# Reading Habits, Socioeconomic Conditions, Occupational Aspiration and Academic Achievement in Vietnamese Junior High School Students

**Thi-Thu-Hien Le [1], Trung Tran [2] , Thi-Phuong-Thao Trinh [3] , Chi-Thanh Nguyen [1], Thuy-Phuong-Tram Nguyen [4], Thu-Trang Vuong [5,*], Thi-Hanh Vu [6] , Dieu-Quynh Bui [7] , Ha-My Vuong [8], Phuong-Hanh Hoang [7], Minh-Hoang Nguyen [9] , Manh-Toan Ho [10,11,*] and Quan-Hoang Vuong [10,11,12,*]**

[1] Faculty of Pedagogy, University of Education, Vietnam National University, Hanoi 100000, Vietnam; hienltt.1978@gmail.com (T.-T.-H.L.); nchithanh@gmail.com (C.-T.N.)

[2] Faculty of Basic, Vietnam Academy for Ethnic Minorities, Hanoi 100000, Vietnam; trungt1978@gmail.com

[3] Department of Mathematics, Thai Nguyen University of Education, Thai Nguyen 24000, Vietnam; trinhphuongthao@dhsptn.edu.vn

[4] Duc Trong High School, Lam Dong 66000, Vietnam; nguyenthuyphuongtramdt@gmail.com

[5] Sciences Po Paris, 75337 Paris, France

[6] School of Economics and International Business, Foreign Trade University, Hanoi 100000, Vietnam; hanhvt@ftu.edu.vn

[7] National Centre for Sustainable Development of General Education Quality, Vietnam National Institute of Educational Sciences, 101 Tran Hung Dao street, Hoan Kiem district, Hanoi 100000, Vietnam; buidieuquynh2019@gmail.com (D.-Q.B.); hoangphuonghanh.hph@gmail.com (P.-H.H.)

[8] Hanoi Amsterdam High School for the Gifted, Hoang Minh Giam Street, Cau Giay District, Hanoi 100000, Vietnam; vuonghamy2003@gmail.com

[9] Ritsumeikan Asia Pacific University, Beppu, Oita 874-8577, Japan; nmhhg8@gmail.com

[10] Center for Interdisciplinary Social Research, Phenikaa University, Ha Dong district, Hanoi 100803, Vietnam

[11] Faculty of Economics and Finance, Phenikaa University, Ha Dong District, Hanoi 100803, Vietnam

[12] Centre Emile Bernheim, Université Libre de Bruxelles, 1050 Bruxelles, Belgium

[*] Correspondence: thutrang.vuong@sciencespo.fr (T.-T.V.); toan.homanh@phenikaa-uni.edu.vn (M.-T.H.); hoang.vuongquan@phenikaa-uni.edu.vn or qvuong@ulb.ac.be (Q.-H.V.)

**Abstract:** Reading practices play an important role in the learning process of students. Especially in a fast-changing world where knowledge about nature and society is in a constant state of flux, book reading helps students foster skills such as thinking, valuing, adaptability and creativity for sustainable development. This research study used a dataset of 1676 observations of junior high school students from Northern Vietnam to explore students' academic achievement and its association with their reading passion, family socio economic condition, parental education and occupational aspiration. The empirical results show that higher grades in STEM-related subjects are predicted by reading interest ($\beta_{\text{Readbook}} = 0.425$, $p < 0.0001$), with students who love reading books achieve higher score than those who take no interest in books. Remarkably, the education level of the mother strongly enhances academic performance, with $\beta = 0.721$ ($p < 0.0001$) in cases of mother having a university diploma or higher. Students coming from wealthy families are more likely to buy books whereas borrowing from the library is the main source of books for students who grow up in not-rich families. However, even among wealthy families, investment into buying books still rely more on personal interest, despite the aforementioned educational benefits of book reading, as evidenced by an over 7 percentage point disparity between the likelihood of purchasing books among wealthy-family students who took an interest in reading (45%) versus students of the same background who did not like to read (38.7%). The results present implications for education policy making with a vision towards United Nations' Sustainable Development Goal 4: Quality Education.

**Keywords:** junior high school students; STEM; reading practices; occupational aspiration; parental influence; socioeconomic background; academic achievement; Vietnam; quality education; sustainable development goal 4

## 1. Introduction

### 1.1. A Brief Overview

Among the different pillars of sustainable development, including equitable economic development and environmental conservation, human capital is considered the epicenter that pulls together all the resources of a nation. The success of a country cannot be divorced from the capacity of its labor force to thrive and adapt to their own needs and societal demands [1]. The emergence of Taiwan, Hong Kong and Singapore as major exporters of advanced products has underlined the significance of human capital investment and acquisition of new capabilities [2]. The same message is proposed by the Boston Consulting Group [3] that by 2030, most countries will face labor shortages, and the key element to top economies in the world in this coming era is human capital.

Over half a century ago, Nobel laureate economist Gary Becker stated that the driving force for lifting labor productivity lies in education [4]. This is because education raises social aspiration and enables people to work towards the qualitative increase of living standards and life satisfaction. Educational attainment level is a key indicator which is associated with employment rates and social equity in a country [1]. The role of education in today's ever-changing world has become even more relevant as its ultimate goal has switched from transmission of knowledge to development of academic and vocational skills in order to extend the capacities of learners. It is commonly accepted that teaching literacy constitutes one of the main goals of education. By extension, self-education and lifelong learning in large part concern the skill of reading comprehension. It is indeed a means for acquiring new knowledge and skills through processing information in textual form, via various mediums including, but not limited to, newspapers, books, and technological devices [5]. A regular habit of reading helps develop a logical thinking mind and craft new ideas by constantly constructing meaning and gaining information from printed text [6]. More importantly, in the contemporary context of global radical technological advancement, language competency is the major tool for learners to adapt to the unprecedented changes and master necessary skills for the 21st century world [7–9]. Studies have suggested that avid readers have higher propensity for critical thinking [10] and self-direction skills [11], which are associated with more self-awareness, clearer life goals and higher achievements in adulthood [12]. Reading proficiency has also been found to be the predicting factor for academic performance, including Science, Technology, Engineering and Mathematics (STEM) subjects, as well as educational attainment at secondary level [13,14].

More recently, as the contribution of science, technology and innovation (STI) to sustainability is well recognized worldwide, investment in STEM education is the key to address issues of shortage and mismatch between skills to job demands in the long run. The United States was one of the pioneers focusing on the STEM education to save their future workers from job loss [15]. Concerns about advancing STEM education to improve the workforce capacity has then quickly become prevalent among educators and policymakers in developed as well as emergent countries such as England, Germany, Japan, Korea and Israel [16–19]. Definitions of STEM education as well as its theoretical concepts and frameworks have been widely researched and developed. Attributes of STEM have their roots in theories and ideas about learning process such as systems thinking [20], situated cognition theory [21–23], constructivism [24], or goal orientation theory [24]. Essentially, all these understandings about STEM share some common emphases: First, the teaching and learning of STEM take place under a combination of different domains, particularly highlighting Mathematics and Science and their connections. This interdisciplinary approach is necessary for developing systematic thinking.

Learning occurs when students take a holistic approach to explore new ideas/problems by connecting different parts to form a whole picture. Second, the learning activities in a STEM class should revolve around an authentic context. Understanding how the outcome skills and knowledge are relevant to real life situations is as important as the learning process itself. Third, STEM activities should be planned and organised from a learner-centred approach which incorporates engineering design process. Learning is optimised when learners are allowed to determine the expected outcomes for themselves and have control over the progress towards the outcomes in which they make sense of their personal learning experiences through problem solving as instructed by teachers. The extent to which sustainable development goals can be achieved through STI relies on the capacity of a system to provide people equal opportunity of access to lifelong education. This is the grounding platform for progression of human consciousness and sustainable development of a nation.

Research into the role of reading as a means for sustainable education, particularly in STEM subjects at the level of secondary education, would perhaps be beneficial in improving the efficiency of the education system in upscaling the quality of the labor force. This goal is even more critical for developing countries in order to avoid the "middle-income trap" where rising labor cost and skill shortages lead to recession in foreign investment [25]. However, most existing studies on reading behaviors and attitudes and academic aptitudes have been conducted in the context of educational systems in the English-speaking world, and largely in developed countries [26–31]. When evaluating the extant literature, one must be aware of the different practices in other cultural spheres [32–34], as well as the reality of social disparities in developing countries [35–38]. More specifically, the association between reading practices and STEM learning results at secondary level of education is generally understudied even in high income economies. It is thus crucial to investigate the reading interests and habits of young people in relations to their academic performance and other capabilities, especially with a view to the SDG4 (https://sustainabledevelopment.un.org/sdg4).

This study contributes to the current literature with insights from Vietnam, a lower middle-income country [39,40]. In particular, findings based on a 1676-observation dataset of Vietnamese junior high school students regarding the associations between their reading practices, access to print materials and performance in STEM subjects are examined. The effects of demographic factors and future career aspirations (as a measure of clarity of future life goals) on students' academic (STEM) performance are also considered. Implications of the analyses of results in this study are of interest to parents, teachers, educators as well as policy-makers in nations with similar socioeconomic context.

### 1.2. Literature Review

#### 1.2.1. STEM Learning Material and Performance in Scientific Subjects

Investigations of time spent on reading books, especially books related to schooling subjects showed a positive effect on academic achievement. For example, result from a survey with 8th grade students in the United States indicated that those who spend more time to read science-related materials got higher grades in scientific subjects in school. Remarkably, this investigation presented a strong correlation of subject attainment with the frequency of reading science related books [41]. Moreover, reading more science books outside of schools has shown positive effects on academic performance. A longitudinal United States-based study on 6th grade students has found that learners who read additional books about science outside of schools perform better in both reading and scientific subjects, compared to their counterparts who do not [42]. Reading books can also potentially help struggling pupils in improving their reading comprehension [27].

Another focus in literature of science related reading was STEM activities outside school. Out-of-school STEM activities has been proven to inspire students [43] and enhance their interest in scientific subjects, a motivation to learn more about science and a better perceptiveness of the world of science [44]. Both nonfiction science and science fiction books are highly appropriate materials for out-of-school reading activities because it can stimulate interests on chemistry and physics among

students as early as from middle school [45]. In particular, out-of-school activities that are well structured by levels of informal scientific programs may even steer students' occupational orientation towards natural sciences [43].

As such, it could be said that in STEM-related extracurricular activities are worth being considered when promoting natural science subjects. Such activities could be as simple as reading on STEM subjects. In fact, reading STEM materials presents a significant influence on readers' language comprehension. Earlier studies conducted by Corvette and his colleagues showed outperformance in comprehension, writing and vocabulary of science among students who, in addition to taking part in hands-on activities, have read scientific texts [46,47]. Another investigation led by Cervetti demonstrated a better information recall from reviewed text of students who read conceptually coherent science paragraphs compared to peers who read unassociated materials [48]. By reading texts related to STEM fields, students have the chance to acquire STEM fields-related vocabulary and use them as a tool to reflect their own understanding, which support the development of technical and theoretical ideas in learners [49]. Interestingly, for student themselves, improving language skills is a reason to read non–academic books outside of school [31].

### 1.2.2. Preference of Reading Topic and Reading Intensity

Given that the activity of reading showed positive influence in students' academic performance, researchers are also interested in favorite reading materials among today's adolescents as well as why those kinds of materials are preferred. The first remarkable investigation might be mentioned conducted by Moje and her colleagues (by interviews and observations) on the content of the reading material and the frequency of reading outside of schools, among youths aged 11 to 17 from an urban community [50]. The result presents a significant relationship between novel-reading regularity and academic performance. Interestingly, most young participants did not like to read about science outside of school [50]. This phenomenon has been observed in other studies, some of which claim that student's interest in scientific information text is developed in relation to other domains such as the interest and reading culture of the family or the career of parents [26,51]. However, a number of investigations conducted in developed countries concluded that scoring higher in tests and examinations is a reason for upper primary school students to read books frequently (with 62% of responses) [31], for college learners to spend more time to read scientific articles [52].

On the aspect of time, in this case with respect to reading frequency, an investigation on 5th-graders in the U.K. and their reading habits at home showed that 51% of students read story books (excluding textbooks) daily while 35% of participants said they only read a few times a week. In regards to school books, nearly half of participants (45%) said that they have never read a textbook at home and 23% of students reported that they read such books "a few times a week" [53]. Moreover, the finding from 288 3rd grade students in the US reveals that both males and females spend more time reading adventure stories because they find these materials relaxing and enjoyable [54]. Pleasure seems to be the most significant motivation for youth to spend time for reading [50,55].

Preferred types of reading materials which inspire school age children to read outside schools were also examined by various researchers. Interestingly, most of reviewed literature reveals the correlation of students reading frequency and non-academic texts. A study with the participation of 5th grade children coming from various populations (in terms of SES and ethnicity) found that they spend more time reading comic books and magazines outside of school [56]. Other investigations in developed contexts reported that comic and magazines were also popular choices for both primary and secondary school English students [57] and for primary students in Singapore at home [31]. For urban adolescents, magazines seem to be the most attractive reading material (with 68% of males and 76% of females). The majority (69%) of 1340 middle school students in the U.S. said that they read more than two books per month at home [58]. Additionally, the second favorite reading material among youths was comic books or books about sports (by boys) and musicians (by girls), all of which amounted to a total of 44%. Only 30% of participants choose the category "books for pleasure" when asked about what they liked

to read. Additionally, short stories and picture books were chosen mostly by high-poverty students' reading for pleasure and reading for pleasure is strongly correlated with reading frequency [50].

### 1.2.3. Socio-Economic Status and Access to Books

Studies have identified the two most common sources of books for children: purchased sources (bookstores, subscriptions, parental purchase) and borrowed sources (classrooms, schools and public libraries). However, the popularity of these sources varied across studies. Earlier research confirmed school libraries as the main access to books [59–62] whereas more recent findings highlighted the dominance of purchased sources [63]. Furthermore, Worthy and colleagues also found that learners eligible for free lunch (i.e., those from low income families) were more likely to borrow books from libraries than to buy them [63]. Apart from financial constraints, one explanation is that fewer books are available for purchase and less bookstores can be found in high poverty neighborhoods than in middle-class ones [64]. This could account for the gap in reading frequency and proficiency between students in low-income and high income groups during the summer when they do not have access to school libraries [65].

An international investigation on family SES and attitudes toward reading among 4th grade students in 33 countries conducted by Chiu and Chow reported that SES factors and home educational resources were strongly related to learners reading scores. The reading scores of higher SES students are higher in comparison with those of lower SES groups, the result was explained by more books being available in richer homes and more favorable attitudes toward reading presented among higher SES parents [66]. In contrary, children from poor or low SES families seem to be poor readers in later education level due to the lack of reading resources at home and lower family literacy [56,67].

Children from higher socio-economic families show more interest in reading due to having more opportunities to read. According to a number of researchers [68–70], families with higher level of income, families with more money, better job status and educational level tend to have more books at home, which can encourage children to start reading and to read on a regular basis. This evidence was supported by later research findings [66,71]. Vastly different book genres at home also means that children have more reading choices, which stimulates their interest to develop frequent reading habit [72]. On the contrary, children from less favorable socioeconomic backgrounds tend to watch more television [56]; they also have to focus more on helping their parents do household chores, or even earn money, rather than spending time on reading [73]. In additon, low income parents did not purchase books due to the lack of money [74]. Several studies have also found that reading is of less enjoyment for children of lower socio-economic backgrounds than those from more advantageous social class [75]. Statistics from PISA 2009 of Australia reported that 33% of students of the lowest SES quartile claimed they did not read for enjoyment, while 17% of the highest SES quartile gave the same answer [76]. Other than the lower level of exposure, it seemed that the usually lesser amount of parental attention in relations to reading habits [77], which was associated to lower socioeconomic backgrounds, played a role in forming reading enjoyment among children. Encouraging children to read seems to be a tough task for parents who come from low SES homes, whereas reading is seen as entertaining among children from higher SES [78]. In fact, the latter often considers reading as a source of pleasure, and they enjoy reading, and see the values of reading. School-age learners are more likely to read books that they find interesting, and enjoy reading thanks to encouragement from their family [79]. Low SES students lack not only the access to books [80], but also this familial encouragement.

In order to address this issue and provide support to disadvantaged children, book give-away programs have been implemented, attracting a large number of research studies examining their effectiveness. Particularly, scientists have found that being given books for ownership is more influential to children than borrowing, even more so if they are allowed to select books of their preference [81]. In other words, the gap in summer reading between different socio-economic groups can be shrunken by simply supplying children with books that they wish to have and can read [82,83]. Other results from the National Literacy Trust's first annual survey have associated book ownership with increased

interest in reading, reading frequency, reading length, number of books read in a month and number of books available at home. The researchers also found that for school-age students who own more books are more likely to be white girls from higher income families [84]. It is therefore reasonable to expect solid links between socio-economic and demographic (namely gender and ethnicity) backgrounds, the accessibility to reading materials, on the one hand, and children's reading habits, on the other hand.

### 1.2.4. Diversity of Book Genres and Reading Habits

The improvement of regular reading habits of students is strongly influenced by reading culture both in families and schools. In school context, both teaching staff and learners need to access various reading materials to improve the teaching and learning quality in the classrooms [74].

The diversity of books in school libraries has also been examined in relation to the reading habits of students and their attitude towards reading. Research studies in both low and high contexts have shown that vastly different books in school and classroom inspire students to read in and outside school. According to a study based in Botswana, the relationship of reading habits in elementary school pupils and the availability of reading materials in schools have also been examined and confirmed to be positive [85]. An impressive result from the project known as "Book Flood" that aims to increase the number of reading materials in schools in Fiji, Singapore and Sri-Lanka (from 100–200 books delivered to each primary school participated into the project) show a dramatic improvement in reading, and other language skills as well as the positive changes of children's attitudes towards reading [74]. Other investigation also reveals that students are more interested in reading if they access or find the desired books [56,86] and even, the time spend by US urban adolescents to read could be reduced (from 6 days to 2 days per week) due to the lack of wanted books [87]. The investment in school libraries with a variety of reading materials should be increased in order to enhance learners' reading and achievements [88].

Among the factors indicated as deterrent to school-age learners' reading habits, inadequate availability of reading materials in schools was rated more highly by surveyed participants [85]. His investigation on reading habit and books availability in primary schools presents that children's reading habit cannot be developed in the schools without a sufficient amount of books available in their libraries. With the fact that most of books in schools are textbooks and teachers' notes, it can explain why more than half of children (53.3%) spend 1-2 h per day to read, the result also shows that some learners read for less than an hour a day. Moreover, the lack of engaging leisure reading materials is one possible reason only 25% of Singaporean young children borrow books from their school library [31].

A wider variety of genre, authors from classic to contemporary literature that allow students to read in and out of schools is considered as a worth reading resource which provide readers opportunities to travel to other worlds and possessed different existing cultures had been claimed on students reading examinations. The richness of books in schools and homes were indicated as the construction of reading identities and "finding" and "regular daily picking up" books that allowed Singaporean school boys to score high in reading achievement as global literate citizens [89]. Students who spend more time on reading favorite comics books claim that they get their reading materials primarily from the school libraries (71%) and the classroom (53%) [58]. By this result, school library seems to be primary to enhancing students' reading.

### 1.2.5. Parent's Education and Career and Student's Academic Performance and Occupational Aspiration

Some researchers proposed that parents of higher educational backgrounds are more aware of the benefits of reading and thus tend to provide their children with more opportunity to read [90] and their children gained higher score in scientific subjects compared to their peers [41]. It is also worth noting that mothers play a more important role in forming children's reading habits at home than fathers since they spend more time on reading, teaching and encouraging their children to read more frequently

than fathers [57,91]. Moreover, well- educated parents seem to encourage their children to pick up reading more [92,93] to procure additional cultural capital that will strongly support their kids [94]. A study on 11th–12th grade students in the US reveals that students whose parents appreciate the values of STEM fields get higher scores in math and science [95]. Additionally, after high school, a number of those students chose to enrollment in college STEM major or present interest in STEM related fields. This seems to be closely linked to the parents' active efforts at encouraging their children to learn about STEM fields, especially at early stages of schooling [45,96].

High scores in Math and Science and participation in STEM competitions are proven to be positively linked to occupational orientations towards STEM fields regardless of SES [43,97,98]. A data presented by Wang [97] showed significant direct effect of 12th-grade math achievement on STEM entrance. Impressively, positive association of math achievement at the 12th grade with intention to pursue STEM fields has been found among students from White communities but it was found null for their Asian counterparts [97]. However, other study confirms that there is no different in math performance between White students and Asian students in the U.S. [99]. This study also mentions that the students who reach a well preparedness of math and science in high school are like to be interest in STEM major. Academic preparedness seem to be a strongest predictor for choosing a STEM subject also claimed by Miller et al. [98]. Moreover, with the increasing popularity of STEM-related competitions (IT, technology, robotics, science fair, or others related science) as out-of-school activities, a survey with freshman students from four year and two year American institutions conducted by those authors reveals STEM career are likely to be chosen more by participants who took part in any STEM contest compared with participants who was not involved in any STEM contest. For example, students who participate in a computer or IT contest are 3.72 times more likely to follow a computer science related career after high school than their peers who did not involve in any similar competition [98].

The difference of parents' influence on children's math and science choices has also been examined [100]. While non-significant influence of father' level of education and mother' level of education on STEM career have been found in Miller's survey [98], other investigations showed a positive influence of parents on children' subject choices. Namely, when parents want their children to perform better in math, they tend to provide more material resources as well as time in guiding and encouraging their children to get involved in extra-curricular activities related to math [100,101]. Impressively, parents who have high educational expectation for their kids will likely to provide substantial support and encouragement to the child to gain better academic performance [99]. It is worth noting that parents expect differently from boys and girls, in terms of academic achievements in the subject of math [100].

Some scholars have suggested that SES is a significant predictor of science and math scores, based on evidences such as surveys on high school students attending public and private schools in America [93,101]. According to these results, higher SES and higher level of education of parents were considered two significant predictors of better 10th grade math score. In the same vein, English students who chose and science and math as their study focus and achieved high results often came from advantageous SES backgrounds. The majority of higher education students in the U.K. who chose scientific disciplines, especially natural sciences, come from advantageous SES groups [102]. The suggestion was supported by more detailed studies, according to which students whose parents work in agriculture or manual labor and/or had a low educational level tend not to report STEM related occupations. One of the possible explanations stated by the authors was the student's awareness of the variety of career options; namely, that students from wealthier backgrounds are more aware of the range of occupational choices they could make [103].

Additionally, the parents' interest in STEM fields also must not be ignored due to its influence on children's future career choice. Students with an interest in science are significantly more likely to claim that their parents are interested in science, than students who present little to no interest in science [51]. However, STEM-related career suggestions made by parents are also often influenced by gender-based stereotypes; for example, mothers are likely to guide their girls to choose non-scientific or

humanities-related careers while both mothers and fathers tend to recommend careers related to physics or other "hard sciences" for boys to follow [104]. It is worthy to note that mothers' predictions and beliefs in an adolescent's success in math and science seem to be a strong predictor to the adolescent's actual choice of discipline to follow after high school [104]. Finally, on the association between gender and interest for scientific disciplines, there is little consensus in the literature. Many studies found that females stated lower level of interest in the fields related to STEM such as medical science, chemistry, especially engineering than males [105,106], whereas others said there was no significant link between gender and orientation towards any of the STEM careers [103]. That being said, representation in STEM competitions was found to be the same between males and females; females, in fact, performed slightly better than males [43,105].

Globally speaking, students who have more opportunities to be involved in science-related context, STEM activities or interest in Math and Science are more likely to select a STEM-related occupation [43]. Especially, students from high SES, who receive more attention and cultural investment into STEM from their parents at early stage see STEM career is a good field [45].

## 1.3. Concepts and Notions

The following notions in our conceptual framework are necessary for a thorough understanding of the study.

The activity of *reading* is the focal point of the study. Reading should be understood in its most encompassing [107] in fact, students were not restricted from any interpretation 'reading' while filling the survey. As such, reading materials could be of any genres (i.e., including comic books, magazines, etc.) and any mediums (i.e., including e-books, digital newspaper, etc.).

*Reading interest* refers to the response to a question in the original survey: "Do you like to read?" This is a self-reported indicator of whether or not respondents take an interest in the activity of reading. The measure is represented by the dichotomous variable "Readbook".

We often employ the term *academic achievement*, or in more concrete cases *average score*, to refer to the respondent's average score in their most recent 45-min tests in the four separate subjects of Mathematics, Physics, Chemistry and Biology. Tests in Vietnam are graded from 0 to 10, to the second decimal digit. Average scores in Vietnam can be calculated down to the third decimal digit [107]. In the survey, students did the calculation of their scores themselves with the help of their teachers. This indicator is coded as *APS45*.

*Occupational aspiration* corresponds to the question "Which job would you like to have in the future?". The question was open; students could name any occupation that came to mind, all of which were recorded. For the specific scope of this paper, however, we have decided to only consider two categories of students: those who were able to name a dream job and those who were not. We coded this as variable *FutureJob* in our dataset.

## 1.4. Research Questions and Hypotheses

*RQ1*: Does the interest in reading books and favorite types of book affect the average score of students' 45-min tests of math, physics, chemistry and biology?

*RQ2*: Is there a correlation between reading interest, type of books and the amount of time spent reading social sciences and humanities books versus time spent reading natural sciences books?

*RQ3*: How do reading habits and financial status of the student's household affect students' book sources?

*RQ4*: How do reading habits and the students' evaluation of the classroom bookcase influence the amount of time spent on reading natural sciences books and social sciences and humanities books?

*RQ5*: Do the presence of an occupational aspiration and the level of education of parents affect student's 45-min tests of math, physics and chemistry?

The hypotheses correspond to the research questions as follows:

**Hypothesis 1 (H1).** *Reading habits and interest in natural sciences books positively predict the average score of students' 45-min tests of math, physics, chemistry and biology*

**Hypothesis 2 (H2).** *Favorite type of books and interest of reading books are statistically associated with the amount of time spent on reading natural science books and on reading social science books.*

**Hypothesis 3 (H3).** *Interest of reading books and the financial status of students' family have a statistically significant influence on the source of book supply*

**Hypothesis 4 (H4).** *Interest of reading books and the variety of bookcase in the classroom would be positively associated with the likelihood of reading types of books more than 30 min a day.*

**Hypothesis 5 (H5).** *The future job, education level of fathers and mothers are positively correlated with the average score of students' 45-min tests of math, physics and chemistry.*

## 2. Materials and Methods

### 2.1. Materials

#### 2.1.1. Dataset

From December 2017 to January 2018, Vuong and Associates office conducted a survey entitled "Studying reading habits and preference of junior high school students in Vietnam". A questionnaire has been designed specifically for the survey, consisting of 26 multiple-choice questions as a majority as well as 3 open questions. These questionnaires were then sent out to junior high schools in the northern Vietnamese province of Ninh Binh. Junior high school students, aged 11 to 15 (from grade 6 to grade 9), were randomly selected to respond to the questionnaires directly, in written form, after having received thorough explanations from their instructors to ensure accuracy and validity of the records. The dataset employed in this paper is a result of the aforementioned survey, consisting of 1676 observations.

#### 2.1.2. Variables

Since we are interested in examining determinants of average scores in math, physics, chemistry and biology of students' 45 min tests, of the amount of time spent on reading natural science books and social and humanities books and of the source of book supply, the following variables were respectively analyzed as the dependent variables:

- *APS45*: the average score of the most recent 45-min tests in mathematics, physics, chemistry and biology. The variable is treated as continuous;
- *TimeSci* and *TimeSoc*: time spent daily reading natural sciences books and social sciences and humanities books, respectively, and self-reported. These variables both consists of two categories: under 30 min ('less30') and 30 min or over ('g30');
- *Source*: the main source of supply from which students obtain books. This was recorded through a multiple-choice question, which the respondent answered by choosing only 1 out of 3 options: borrowing from friends or libraries ('borrow'), using their own or their parents' money ('buy'), or receiving books as gifts or rewards ('gift').

The analyses would also contain the following independent variables:

- *Readbook*: whether or not a student is interested in the activity of reading, with two answers: 'yes' and 'no';

- *Topic*: student's most preferred reading topic. The student was required to choose only one from a list of options, consisting of: math—physics ('math.phy'), literature ('literality'), foreign languages ('language'), natural sciences, chemistry, and biology ('nat.chem.bio'), history and geography ('his.geo'), information technology ('tech'). This variable was not directly used in the analyses, but recoded into variable *Topicgr*;
- *Topicgr*: this variable is a recode of variable *Topic*. It collapses the categories of variable *Topic* into 2 groups: Group 1 ('gr1') consists of 'math.phy' and 'nat.chem.bio' and group 2 ('gr2') consists of 'literality', 'language', 'his.geo', 'tech' and notans;
- *Bookcase*: student's evaluation of the common bookcase in the classroom. The student responded by choosing only 1 out of 4 options: diverse and interesting ('a'), missing good titles ('b'), lacking books ('c') and no bookcase ('d'). This variable was not used directly but recoded into variable *Bookcasegr* for analysis;
- *Bookcasegr*: this variable has been recorded from *Bookcase*, grouping categories 'a' and 'b' into 'variety', and 'c' and 'd' into 'novariety';
- *Typebook*: student's preferred book genre if ever being gifted a book, excluding textbooks. The student was required to choose only one of the following four options: Novel ('a'), Biography ('b'), Popular Science ('c'), Arts ('d'), Vocational instruction ('e'), and Other ('f'). This variable was not used directly, but recoded into *Typebookgr* for further analysis;
- *Typebookgr*: recoded variable from *Typebook*, grouping 'a', 'b' and 'd' into 'gr1' and 'c', 'e' and 'f' into 'gr2'. Group 1 as represented by 'gr1' consists of novels, biographies and arts, which are characterized by the story of human and human relationships. This group of book type is either fiction books or social sciences and humanities books. Group 2 as represented by 'gr2' is comprised of the genres popular science and vocational instruction (and others), which could be considered non-fiction or natural science books because they mainly focus on practical knowledge and vocational skills;
- *EcoStt*: self-reported financial status of the student's household. The student was required to choose one option from a list of three: rich ('rich'), medium ('med') and poor ('poor'). This variable was not directly used but recoded into *EcoStt2* for analyses;
- *EcoStt2*: recoded from *EcoStt*, this variable collapses 'rich' and 'med' into 'notpoor'; 'poor' remains its own category;
- *Readstory*: whether or not the respondent's parents read books for them, with two answers: 'yes' and 'no';
- *FutureJob*: the presence of concrete future occupational aspirations, self-reported. Respondents who are able to give a concrete answer were coded as 'define'; the rest were 'undefine'.
- *EduFat and EduMot:* refer to academic level of the father and the mother. Academic level contains 4 items: Under high school ('UnderHi'), high school ('Hi'), Undergraduate ('Uni'), Graduate school ('PostGrad').

### 2.2. Methods

Responses were entered into a Microsoft Excel spreadsheet; the XLS file was then converted into CSV to be treated in R. Baseline-category logit (BCL) model were used to analyze the relation between pairs of variables, or between a dependent variable and multiple predictor variables [108].

General linear model (GLM) was utilized to explore the association between independent variables x a continuous dependent variable Y. The equation obtained can be formulated as follows:

$$Y = \beta_0 + \beta_i x_i$$

in which $\beta_0$ is the intercept and $\beta_i$ is the coefficient corresponding to each independent variable.

Binomial and multinomial logistic regression was employed to estimate the probability of a category of dependent variable Y against different values of independent variables x, in order

to assess how the response variable varies when the predictor variables change. Through these regressions, estimate coefficients were obtained and later used to compute conditional probabilities.

The general equation of the logistic regression model is as follows:

$$\ln \frac{\pi_j(\mathbf{x})}{\pi_J(\mathbf{x})} = \alpha_j + \beta'_j \mathbf{x}, j = 1, \dots, J - 1.$$

in which $\mathbf{x}$ is the independent variable; and $\pi_j(x) = P(Y = j|x)$ is the corresponding probability. $\pi_j = P(Y_{ij} = 1)$ with Y being the dependent variable.

The probability of the values of the dependent variable is calculated as follows:

$$\pi_j(\mathbf{x}) = \frac{\exp\left(\alpha_j + \beta'_j \mathbf{x}\right)}{1 + \sum_{h-1}^{J-1} \exp\left(\alpha_j + \beta'_j \mathbf{x}\right)}$$

with $\sum_j \pi_j(\mathbf{x}) = 1$; $\alpha_J = 0$ and $\beta_J = 0$; in which $n$ is the number of observations in the sample, $j$ are the categorical values of an observation $i$, and $h$ is the number of rows in matrix $\mathbf{X}_i$.

The statistical significances of all models in this paper were assessed based on $z$-value and $p$-value, with $p < 0.1$ as the threshold for statistical significance.

## 3. Results

### 3.1. Descriptive Statistics

The data shows that students are distributed rather evenly between the four school grades, with sixth graders taking the largest share in the sample (~28%). 95% of student at 6th grade responded that they are interested in books whereas only 82.3% of 9th grade students like reading books. The number of male and female students is relatively similar. Students mainly access books by borrowing from friends or libraries (approximately 61%). About 37% of students purchase books, and a very small amount of students receive books as gifts or rewards (2.57%) [109]. Regarding the financial status of students' households, the majority of students come from families with a medium level (82.33%); a small number of students report to come from rich and poor families (11.1% and 6.57% respectively). In particularly, within the grade 6 group, nearly 76.7% respondents grew up in medium economic conditions while 16.7% reported that their household was rich.

Notably, most of the students who were interviewed had a concrete idea of their preferred future job (99.34%). In the dataset, students mentioned 104 different jobs among which "doctor" is the most favored (18.14%), closely followed by "police officer" (13.15%) and teaching profession (11.29%). The remaining jobs are evenly distributed with a low percentage. Distribution of categorical variables that were used is shown in Table 1.

**Table 1.** Distribution table of some categorical variables.

| Code Name | Explanation | Items | Frequency | Proportion |
|---|---|---|---|---|
| Grade | Current grade | Grade 6 | 467 | 27.86% |
| | | Grade 7 | 443 | 26.43% |
| | | Grade 8 | 410 | 24.46% |
| | | Grade 9 | 356 | 21.24% |
| Sex | Biological gender | Male | 853 | 50.89% |
| | | Female | 823 | 49.11% |
| Source | Main source of supply for books | Buy | 617 | 36.81% |
| | | Borrow | 1016 | 60.62% |
| | | Gift | 43 | 2.57% |
| Ecostt | Economic condition of the family | Rich | 186 | 11.10% |
| | | Medium | 1380 | 82.33% |
| | | Poor | 110 | 6.57% |
| Topicgr | Answer to question: "Which reading topic do you prefer?" | Group 1 ('math.phy' and 'nat.chem.bio') | 578 | 34.49% |
| | | Group 2 ('literality', 'language', 'his.geo', 'tech' and notans) | 1098 | 65.51% |
| Bookcasegr | Evaluating classrooms' public bookshelves | Group 1 (diverse and interesting ('a'), missing good titles ('b') into 'variety' | 1092 | 65.15 |
| | | Group 2 (lacking books ('c') and no bookcase ('d') into 'novariety' | 584 | 34.85 |
| TimeSci | Time per day spent on natural science books | Under 30 min | 846 | 50.48% |
| | | 30 min or over | 830 | 49.52% |
| TimeSoc | Time per day spent on social science—literary books | Under 30 min | 1065 | 63.54% |
| | | 30 min or over | 611 | 36.46% |
| EduFat | Academic level | Under high school | 1068 | 65.6% |
| | | High school | 433 | 26.6% |
| | | Undergraduate | 97 | 5.96% |
| | | Graduate school | 30 | 1.84% |
| EduMot | Academic level | Under high school | 1033 | 62.34% |
| | | High school | 435 | 26.25% |
| | | Undergraduate | 156 | 9.41% |
| | | Graduate school | 33 | 1.99% |
| Readbook | Answer to question: "Do you like reading books?" | Yes | 1512 | 90.21% |
| | | No | 164 | 9.79% |
| Futurejob | Answer to question: "In the future, which job do you like most?" | Define | 1665 | 99.34% |
| | | Undefine | 11 | 0.66% |

Table 2 is the information for the continuous variable 'APS45'.

**Table 2.** Average score of the most recent 45-min tests of Math, Physics, Chemistry and Biology of students.

| Coded Name | Unit | Explanation | Mean | Min | Max | Standard Deviation |
|---|---|---|---|---|---|---|
| APS45 | Count | Average score of the most recent 45-min tests of Math, Physics, Chemistry and Biology of students. | 19.95 | 2.30 | 9825.00 | 348.01 |

The distribution of average score of the most recent 45-min tests of math, physics, chemistry and biology of students can be seen in Figure 1:

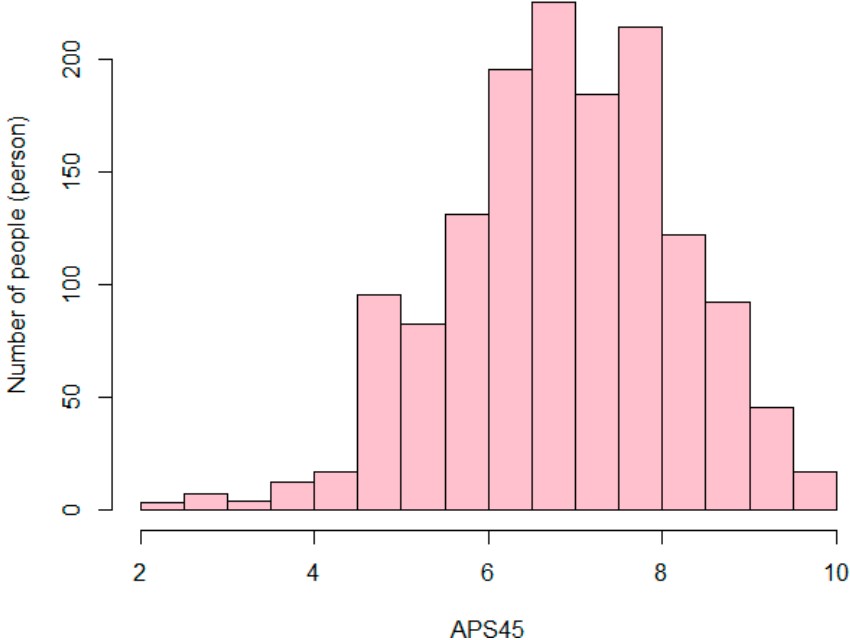

**Figure 1.** Distribution of the average score of the most recent 45-min tests of math, physics, chemistry and biology of students (recreated from [109]).

### 3.2. Regression Results

#### 3.2.1. RQ1–H1

Does the interest in reading books and favorite types of book affect the average score of students' 45-min tests of math, physics, chemistry and biology of students?

We employed GLM estimation with the continuous variable "APS45" as dependent variable against two independent variables "Readbook" and "Topicgr". "Readbook" and "Topicgr" were treated as dichotomous variables. The result is shown in Table 3, which displays that all correlations are statistically significant ($p < 0.0001$). As can be seen that, students who reported an interest in reading book score higher than those who did not, by 0.425 point. Book topic preference was also predictive of grade, in which students who preferred to read about natural sciences (mathematics, physics, chemistry and biology) acquired higher score than students choosing other topics.

**Table 3.** Estimate results of "APS45" by "Readbook" and "Topicgr".

|  | Intercept | "Readbook" | "Topicgr" |
|---|---|---|---|
|  |  | "yes" | "gr2" |
| **"APS45"** | $\beta_0$ 6.808 *** [53.896] | $\beta_1$ 0.425 *** [3.637] | $\beta_2$ −0.265 *** [−3.650] |

Significance codes: 0 '***' 0.001 '**' 0.01 '*' 0.05; t-value in [square brackets]; baseline category for: "Readbook" = "no", "Topicgr" = "gr1". Null deviance: 2469.5 on 1443 degrees of freedom. Residual deviance: 2418.4 on 1441 degrees of freedom. AIC: 4850.6.

Observing that "Topicgr" at "gr2" yields a negative coefficient, it is worth keeping in mind that "gr2" denotes social sciences and humanities books. As such, this result confirms the hypothesis.

#### 3.2.2. RQ2–H2

*Is there a correlation between reading interest, type of books and the amount of time spent reading social sciences and humanities books versus time spent reading natural sciences books?*

To test the relationship between reading interest, favorite types of books and duration of time spent on reading books, we ran two models employing logistic regression, where "TimeSci" and "TimeSoc" are dependent variables and independent variables are "Typebookgr" and "Readbook". The results are presented in Table 4:

**Table 4.** Estimate results of "TimeSci" by "Readbook" and "Topicgr".

|  | Intercept | "Typebookgr" | "Readbook" |
|---|---|---|---|
|  |  | "gr2" | "yes" |
| logit(g30\|less30) | $\beta_0$ <br> −0.924 *** <br> [−4.822] | $\beta_1$ <br> −0.571 *** <br> [−5.610] | $\beta_2$ <br> 1.253 *** <br> [6.475] |

Significance codes: 0 '***' 0.001 '**' 0.01 '*' 0.05; z-value in [square brackets]; baseline category for: "Typebookgr" = "gr1", "Readbook" = "no". Null deviance: 2323.3 on 1675 degrees of freedom. Residual deviance: 2238.4 on 1673 degrees of freedom AIC: 2244.4.

Using estimate coefficients, all probabilities are calculated similarly to the following example, which gave the "TimeSci" = "g30" probability of a student who prefers Group 2 books ("Typebookgr" = "gr2") and reportedly takes an interest in reading ("Readbook" = "yes"):

$$\ln\left(\frac{\pi_{g30}}{\pi_{less30}}\right) = -0.924 - 0.571 \times gr2 + 1.253 \times yes$$

$$\pi_{g30} = \frac{e^{-0.924-0.571\times1+1.253\times1}}{1 + e^{-0.924-0.571\times1+1.253\times1}} = 0.439$$

The probability of "TimeSci" by "Typebookgr" and "Readbook" is described in Table 5 and visualized in Figure 2. Reading interest and favorite type of book could predict the time spent reading natural sciences book. Namely, being interested in reading (Readbook = "yes") and preferring natural sciences books (Typebookgr = "gr2") are predictive Indeed, students who answered yes to "Readbook" and were interested in "gr1" books (see Variables) had the highest likelihood (58.1%) to spend more than 30 min reading per day, while students not having reading book interest only obtained 28.4% likelihood to spend more than 30 min a day reading science book. Students interested in "gr2" books reported much lower likelihoods to spend 30 min reading science books than students interested in "gr1", regardless of whether they take an interest in reading (43.9% among those who reported reading interest; 18.3% among those who reported no reading interest).

**Table 5.** Probabilities of "TimeSci" against "Readbook" and "Typebookgr".

| "TimeSci" | "g30" | | "less30" | |
|---|---|---|---|---|
| "Readbook"\| "Typebookgr" | "yes" | "no" | "yes" | "no" |
| "gr2" | 0.439 | 0.183 | 0.560 | 0.816 |
| "gr1" | 0.581 | 0.284 | 0.418 | 0.715 |

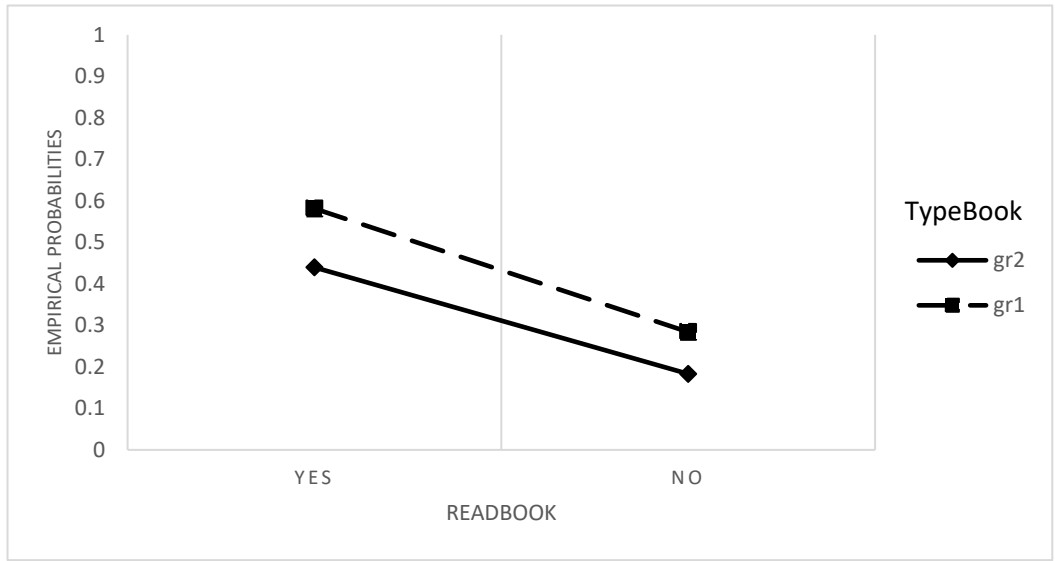

**Figure 2.** Probabilities of "TimeSci" = "g30" against "Readbook" and "Typebookgr".

Figure 2 visualizes the probabilities in all four cross-tabulated categories of the two dichotomous independent variables.

Estimate coefficients from the above Table 6 were employed to calculate probabilities. The following example gave the "TimeSoc" = "g30" probability of a student who prefers Group 2 books ("Typebookgr"="gr2") and reportedly takes an interest in reading ("Readbook"="yes"):

$$\ln\left(\frac{\pi_{g30}}{\pi_{less30}}\right) = -1.583 - 0.266 \times gr2 + 1.232 \times yes$$

$$\pi_{g30} = \frac{e^{-1.583-0.266\times1+1.232\times1}}{1 + e^{-1.583-0.266\times1+1.232\times1}} = 0.350$$

**Table 6.** Estimate results of "TimeSoc" by "Readbook" and "Topicgr".

|  | Intercept | "Typebookgr" | "Readbook" |
|---|---|---|---|
|  |  | "gr2" | "yes" |
|  | $\beta_0$ | $\beta_1$ | $\beta_2$ |
| logit(g30\|less30) | −1.583 *** | −0.266 * | 1.232 *** |
|  | [−7.099] | [−2.534] | [5.502] |

Significance codes: 0 '***' 0.001 '**' 0.01 '*' 0.05; z-value in [square brackets]; baseline category for: "Typebookgr" = "gr1", "Readbook" = "no". Null deviance: 2198.9 on 1675 degrees of freedom. Residual deviance: 2152.5 on 1673 degrees of freedom. AIC: 2158.5.

As for time spent reading social sciences and humanities books, represented by "TimeSoc", probabilities were described in Table 7 and visualized in Figure 3. Results are similar to those obtained in the prior model employing "TimeSci" as a dependent variable. More precisely students having reading book interest also acquired higher probability of spending more time reading social science books, and if their favorite type of book belonged to "gr2", they would be less likely to spend more than 30 min reading social science books. To elaborate, the probability of students reporting to have reading book interest and be interested in "gr2" books was 35%, while that of students reporting to be interested in "gr1" was higher with 41.3%.

**Table 7.** Probabilities of "TimeSoc" against "Readbook" and "Typebook".

| "TimeSoc" | "g30" | | "less30" | |
|---|---|---|---|---|
| "Readbook"\| "Typebookgr" | "yes" | "no" | "yes" | "no" |
| "gr2" | 0.350 | 0.136 | 0.650 | 0.864 |
| "gr1" | 0.413 | 0.170 | 0.587 | 0.830 |

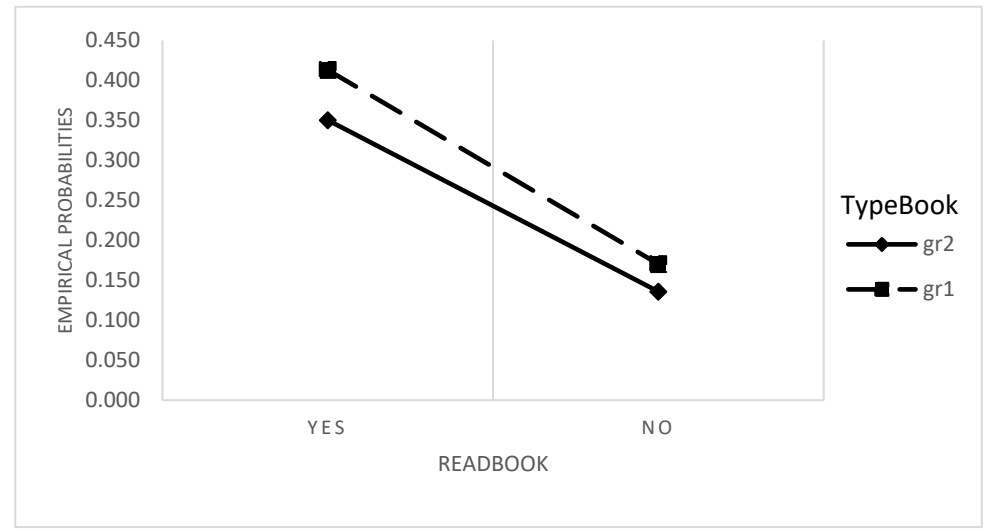

**Figure 3.** Probability of time reading social science books by reading interest and type of book.

### 3.2.3. RQ3–H3

*How do reading habits and financial status of the student's household affect students' book sources?*

Categorical logistic regression analysis was employed to estimate the sources from which students received their books. (For dependent variable "Source" which had three categories, "buy", "borrow", and "gift", we selected "gift" as the base-line reference for the analysis, as shown in Table 8).

**Table 8.** Estimate results of "Source" by "Readbook" and "Ecostt".

| | Intercept | "Readbook" | "EcoStt" |
|---|---|---|---|
| | | "yes" | "rich" |
| | $\beta_0$ | $\beta_1$ | $\beta_2$ |
| logit(buy\|gift) | 2.048 *** | 0.929 * | −0.954 ** |
| | [5.297] | [2.214] | [6.475] |
| logit(borrow\|gift) | 2.719 *** | 0.794 | −1.407 *** |
| | [7.232] | [1.941] | [−3.921] |

Significance codes: 0 '***' 0.001 '**' 0.01 '*' 0.05; t-value in [square brackets]; baseline category for: "Readbook" = "no", "EcoStt" = "not rich". Residual deviance: 2543.517 on 3346 degrees of freedom. Log-likelihood: -1271.758 on 3346 degrees of freedom.

All the coefficients were statistically significant with $p < 0.05$, except for $\beta_1$ of the equation of logit(borrow\|gift) as dependent variable. The latter still had a *p*-value lower than 0.1; as such, we still considered it worthwhile for consideration in evaluating the influence of "Readbbook" on logit(borrow\|gift). The equations of nominal logistic regression were presented as follows:

$$\ln\left(\frac{\pi_{buy}}{\pi_{gift}}\right) = 2.048 + 0.929 \times yes - 0.954 \times rich$$

$$\ln\left(\frac{\pi_{borrow}}{\pi_{gift}}\right) = 2.719 + 0.794 \times yes - 1.407 \times rich$$

where:

$$\pi_{buy} = \frac{e^{2.048+0.929\times1-0.954\times1}}{1 + e^{2.048+0.929\times1-0.954\times1} + e^{2.719+0.794\times1-1.407\times1}} = 0.450$$

$$\pi_{borrow} = \frac{e^{2.719+0.794\times1-1.407\times1}}{1 + e^{2.048+0.929\times1-0.954\times1} + e^{2.719+0.794\times1-1.407\times1}} = 0.489$$

The probability of "Source" is presented in Table 9 and visualized in Figure 4. From the probability results in Table 9, there are several notable findings. First, economic status has a strong influence on the students' means of obtaining books. In reportedly rich families (45% and 38.7% in case of having reading interest or not respectively), students were most likely to buy books and least likely to receive them as gifts (5.9% and 12.9% in case of having reading interest or not respectively), out of all the means of access to books listed in the survey. The same could be said about student from medium families. On the other hand, students not obtaining a wealthy background tend to borrow books with a likelihood of more than 60%.

**Table 9.** Probabilities of "Source" against "EcoStt" and "Readbook".

| "Source" | "buy" | | "borrow" | | "gift" | |
|---|---|---|---|---|---|---|
| "EcoStt"\| "Readbook" | "rich" | "not rich" | "rich" | "not rich" | "rich" | "not rich" |
| "yes" | 0.450 | 0.362 | 0.489 | 0.619 | 0.059 | 0.018 |
| "no" | 0.387 | 0.324 | 0.482 | 0.633 | 0.129 | 0.041 |

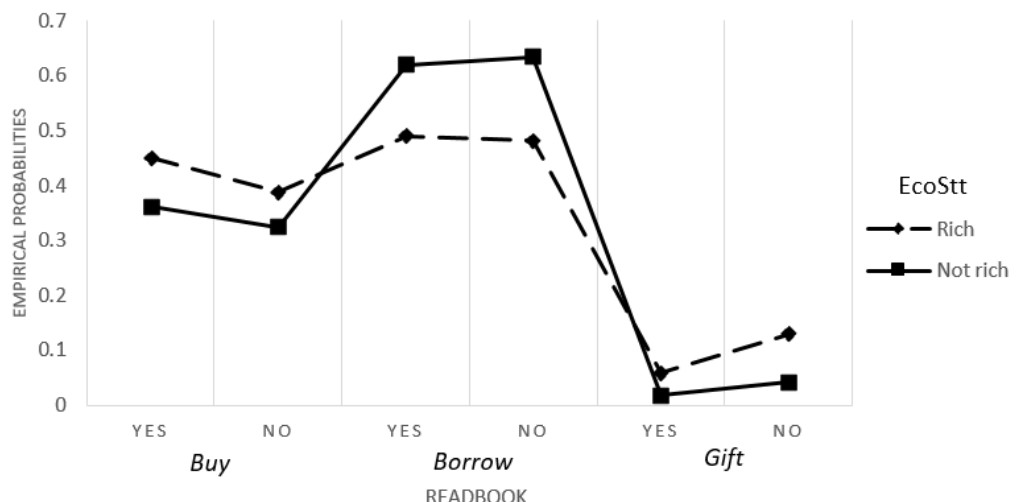

**Figure 4.** Probability of methods of obtaining books by reading interest and family's economic status.

It is worth noting that the effect of reading interest is stronger in predicting means of access to books among students from medium families, compared to students from rich families. In fact, for students from a wealthy background, students who already took an interest in reading are 45% likely to buy books, whereas the probability is 38.7% for those who did not take interest in reading—the gap is 7.5 percentage points. For students from medium families, the gap is only 4.2 percentage points (between 36.2% in case of Readbook = "yes" and 32.4% in case of Readbook = "no").

Second, there was a very small probability that students received books as gifts. Gifting books might be considered as a means to raise reading interest among students, regardless of their backgrounds. The scenario that students obtained the highest probability to received books from gifting was "rich" economic status and "no" reading book interest.

The probabilities are visualized in the following figure.

### 3.2.4. RQ4–H4

*How do reading habits and the students' evaluation of the classroom bookcase influence the amount of time spent on reading natural sciences books and social sciences and humanities books?*

The binary logistic regression analysis of dependent variable "TimeSci" against "Readbook" and "Bookcasegr" was examined and reported in Table 10.

$$\ln\left(\frac{\pi_{g30}}{\pi_{less30}}\right) = -1.254 + 0.220 \times variety + 1.263 \times yes$$

$$\pi_{g30} = \frac{e^{-1.254+0.220\times1+1.263\times1}}{1 + e^{-1.254+0.220\times1+1.263\times1}} = 0.556$$

**Table 10.** Estimate results of "TimeSci" by "Readbook" and "Bookcasegr".

|  | Intercept | "Bookcasegr" | "Readbook" |
|---|---|---|---|
|  |  | "variety" | "yes" |
|  | $\beta_0$ | $\beta_1$ | $\beta_2$ |
| logit(g30\|less30) | −1.254 *** <br> [−6.702] | 0.220 * <br> [2.149] | 1.263 *** <br> [6.556] |

Significance codes: 0 '***' 0.001 '**' 0.01 '*' 0.05; z-value in [square brackets]; baseline category for: "Bookcasegr" = "no variety", "Readbook" = "no". Null deviance: 2323.3 on 1675 degrees of freedom. Residual deviance: 2265.6 on 1673 degrees of freedom. AIC: 2271.6.

The probability of "TimeSci" by "Bookcasegr" and "Readbook" is described in Table 11 and visualized in Figure 5.

**Table 11.** Probabilities of "TimeSci" against "Readbook" and "Bookcase".

| "TimeSci" | "g30" | | "less30" | |
|---|---|---|---|---|
| "Readbook"\| "Bookcasegr" | "yes" | "no" | "yes" | "no" |
| "variety" | 0.556 | 0.262 | 0.443 | 0.737 |
| "no variety" | 0.502 | 0.221 | 0.497 | 0.778 |

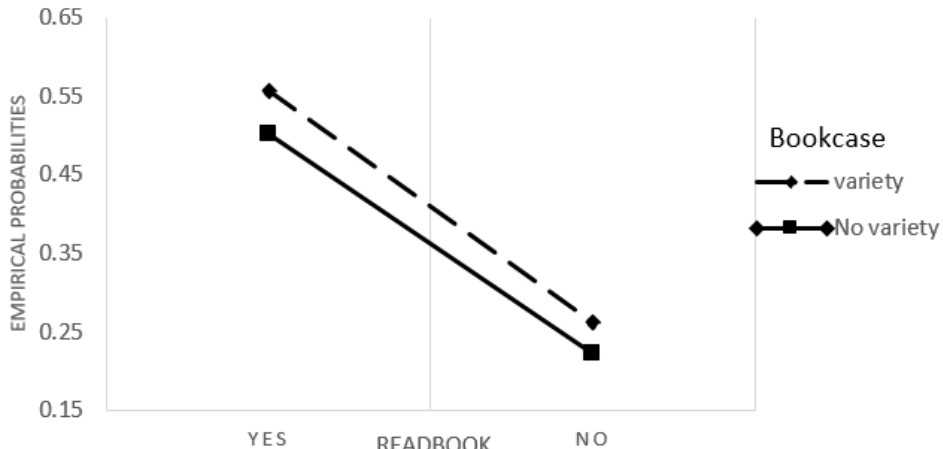

**Figure 5.** Probability of "TimeSci"="g30" by "Readbook" and "Bookcasegr".

The probability of spending more than 30 min per day reading natural sciences books is found to be the highest when the student takes an interest in reading ("Readbook" = "yes") and evaluated the selection of books offered by the common bookcase as varied ("Bookcase" = "variety"). It is worth

noting that reading interest drastically increases the probability of higher daily reading time (around 30 percentage points). Meanwhile, a favorable evaluation of the classroom bookcase only raises this probability by about 5 percentage points. Figure 5 below visualizes the probabilities.

*How do reading habits and the students' evaluation of the classroom bookcase influence the amount of time spent on reading social sciences and humanities books?*

Table 12 reported the results of the binary logistic regression analysis performed on dependent variable "TimeSoc" against "Readbook" and "Bookcasegr".

**Table 12.** Estimate results of "TimeSoc" by "Readbook" and "Bookcasegr".

|  | Intercept | "Bookcasegr" | "Readbook" |
| --- | --- | --- | --- |
|  |  | "variety" | "yes" |
|  | $\beta_0$ | $\beta_1$ | $\beta_2$ |
| logit(g30\|less30) | −1.838 *** | 0.448 *** | 1.198 *** |
|  | [−8.349] | [4.269] | [5.337] |

Significance codes: 0 '***' 0.001 '**' 0.01 '*' 0.05; z-value in [square brackets]; baseline category for: "Bookcasegr" = "no variety", "Readbook" = "no". Null deviance: 2198.9 on 1675 degrees of freedom. Residual deviance: 2140.8 on 1673 degrees of freedom AIC: 2146.8.

The regression analysis can also be expressed by the following equation:

$$\ln\left(\frac{\pi_{g30}}{\pi_{less30}}\right) = -1.838 + 0.448 \times variety + 1.198 \times yes$$

where:

$$\pi_{g30} = \frac{e^{-1.838+0.448\times1+1.198\times1}}{1 + e^{-1.838+0.448\times1+1.198\times1}} = 0.452$$

The probability of "TimeSoc" by "Bookcasegr" and "Readbook" is described in Table 13 and visualized in Figure 6.

**Table 13.** Probabilities of "TimeSoc" against "Readbook" and "Bookcase".

| "TimeSoc" | "g30" |  | "less30" |  |
| --- | --- | --- | --- | --- |
| "Readbook"\| "Bookcasegr" | "yes" | "no" | "yes" | "no" |
| "variety" | 0.452 | 0.199 | 0.548 | 0.801 |
| "no variety" | 0.345 | 0.137 | 0.655 | 0.863 |

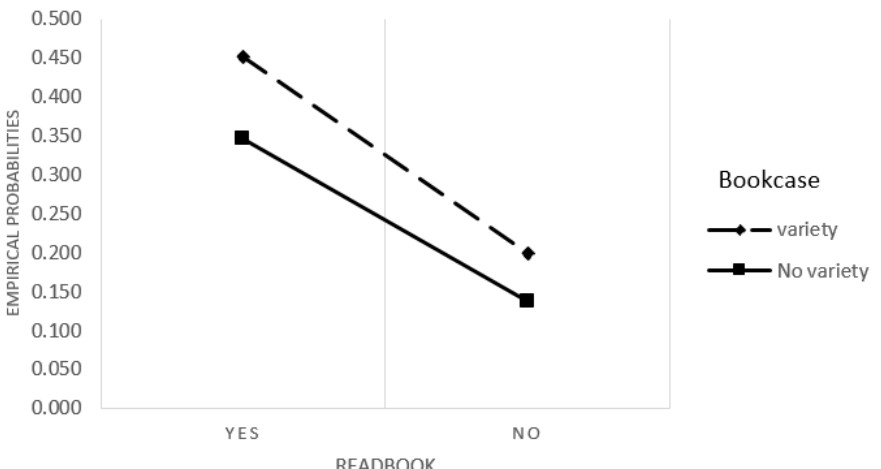

**Figure 6.** Probability of time reading social science books by reading interest and type of classroom's bookcase.

The regression analysis yielded on "TimeSoc" similar results to the previous model. Students, who had an interest in reading and who considered their classroom bookcase as having a considerable variety of books, were the most likely to spend more than 30 min a day reading social sciences and humanities books (45.2%). The likelihood is 34.5% for those who also reported positive reading interest but evaluated the classroom bookcase as unvaried. On the other hand, these probabilities were 19.9% and 13.7% students who answered no to "Readbook", and considered their classroom bookcase to be diverse and not diverse, respectively.

### 3.2.5. RQ5–H5

*Do the presence of an occupational aspiration and the level of education of parents affect student's 45-min tests of math, physics and chemistry?*

We conducted a regression analysis (general linear model) with the dependent variable "APS45" against three independent variables: "FutureJob", "EduFat" and "EduMot". Table 14 presented that all coefficients are statistically significant with $p < 0.05$.

**Table 14.** Estimate results of "APS45" by "FutureJob", "EduFat", and "EduMot".

|  | Intercept | "FutureJob" | "EduFat" | "EduMot" |
|---|---|---|---|---|
|  |  | "Defined" | "Uni" | "Uni" |
| **"APS45"** | $\beta_0$ <br> 6.605 *** <br> [46.790] | $\beta_1$ <br> 0.337 * <br> [2.325] | $\beta_2$ <br> 0.306 * <br> [1.973] | $\beta_3$ <br> 0.721 *** <br> [5.586] |

Significance codes: 0 '***' 0.001 '**' 0.01 '*' 0.05; t-value in [square brackets]; baseline category for: "FutureJob" = "Define", "EduFat" = "EduMot" = "Uni". Null deviance: 2392.2 on 1400 degrees of freedom. Residual deviance: 2270.0 on 1397 degrees of freedom. AIC: 4662.

The regression analysis results can also be expressed in equation form as follows:

$$APS45 = 6.605 + 0.337 \times Define + 0.306 \times Uni_{EduFat} + 0.721 \times Uni_{EduMot}$$

It was found that the presence of an existing occupational aspirations and university-level (or above) education of parents have an enhancing effect on academic achievement. Among three variables, the education level of the mother was the most impactful factor on the score.

## 4. Discussion

### 4.1. Reading Interest, Favorite Books and Time Spent Reading

A vast body of literature has reaffirmed the role of reading on students' process of learning since reading practices help foster students' skills such as talking, thinking, interacting, valuing and believing as well as their ability to expand perceived knowledge about nature and society [110,111]. Students in different grades must step by step learn to perceive the world by reading books which are mainly categorized as natural science books and social sciences-humanities books and students are more likely to read natural science books once they need to seek supplementary knowledge in the field besides their teacher—created lessons. Study results at school in turn motivate students to accumulate knowledge and skills through reading books. Therefore, cultivating reading interest of students is tantamount to students' academic success.

Reading interest of Vietnamese secondary students varies by school grades. 6th grade students are reportedly more likely to be interested in reading books than older students. One of the possible reasons might be that students in higher grades had less time to spend on reading. The more demanding curriculum in higher school grades puts more pressure on students, which takes away time from leisure reading. This is particularly true for students in 9th grade, who usually have to prioritize preparations for the national high school entrance exams [107].

Regarding the correlation between reading interest and academic achievements, our results showed that students who are interested in reading books average higher score of 45-min tests of Math, Physics, Chemistry and Biology (APS45) compared with those who reportedly take no interest in reading. Our finding is in line with the study of Whitten et al. [112] in that there is the close link between pleasure reading and academic success of high school juniors at a rural Southeast Texas high school. Because reading is the fundamental process of learning and getting knowledge [6], the evaluation process through test scores reflects students' ability to perceive intellectual treasure of the mankind.

Naturally, each student is apt to choose favorite books to read when he or she is aware that a type of books provides them with supplementary knowledge which in turn help improve study results of students at school. In the study, we found that student choosing maths, science, chemistry and biology books acquired higher score in scientific subjects than student choosing social science topic. This result supports the finding of Zhihui Fang that students reading more science books are more likely to perform better in science subjects than their peers [42]. The association between being interested in natural sciences books and scoring higher in natural sciences subjects should not come as a surprise. As such, when planning initiatives to develop skills in STEM-related school subjects among students, one should take into consideration not only hands-on activities but also book reading on STEM topics, to enhance the positive effects.

Interesting findings of the influence of time spent on studying and working on academic performance of students are that time spent on working has not had direct impact on colledge students' semester grade point average (SGPA) whereas the interaction between ATC composite score and time spent working significantly influences students' academic achievement [113]. Nonetheless, a big gap in the literature on the association between the amount of time spent on reading and students' academic success should be filled up. One important point to note is that our findings showed positive and insignificant association between the amount of time spent on reading books and students' academic success. As such, the amount of time spent reading—be them natural sciences or social sciences and humanities books—might not be the determinant of academic performance. This could suggest that intensive reading habits do not necessarily translate to immediate and measurable results in school. Whether it is a matter of the schooling system and its evaluation criteria could constitute a subject for future research.

Our data shows that, 6th grade students tend to spend more time on reading natural science books than the olders probably because younger students are under less pressure from schoolwork; therefore reading time gives them more pleasure. On the contrary, most of the 9th grade students spend less than 30 min per day reading social science books. Social science books take more time to read compared with their counterparts and the 9th grade students spend most of their time on studying as their top priority therefore spending less time on reading social science books [96]. Interestingly, favorite types of book determine the larger amount of time spent on reading such books. This implies that a personal interest for such types of book may increase students' energy thus making them more relaxed even with more time spent reading. The question that remains hangs on how this interest forms for individual students.

### 4.2. Family Socioeconomic Status

Socioeconomic background and its relationship with educational achievement of students is one of the enduring issues in educational research [114]. Some researcher argue that students' academic success is positively influnced by the financial condition of their family. It is true that, wealthier parents are able to invest more in education of their sons. Rich parents in many cases spend money on students' learning resources such as books, educational games, laptop as well as on extracurricular courses [115,116].

In our paper, we investigated the association between students' family socioeconomic status and source of book supply. Students in rich families were more likely to buy books and receive them as gifts than students of the other two categories of SES, regardless of reading interest. In addition, students who do not come from a wealthy background are more likely to borrow. The finding implies

that students with a more favorable financial bacground are willing to spend their own money to buy books whereas students in not rich families still obtain books to read by borrowing, perhaps, due to the economic constraints and the shortage of bookstores in the neighborhood [63,64].

This finding is in line with a study on fifteen-year-old students in 9th grade from rural areas of five provinces in western China [117], according to which there exists a positive correlation between the students' family SES index and academic performance. The vital role of family SES on student's academic outcomes is reinforced in a range of educational research [77,118,119]. It is worth noting that family SES has been shown to have a strong correlation with language achievement, more so than in science/math and in general achievement [118]. Additionally, the same study also pointed out that the positive correlation between family SES and academic performance is not always consistent. Nevertheless, our paper contributes further evidence for the relationship between socioeconomic conditions of the household and the student's performance in STEM-related subjects. Students growing in rich families are better financially supported and oftentimes do not have to spend time on chores or helping their parents generate income (farming, small retail businesses, etc.), thus having more time on their study at school as well as to read [56,73].

It should also be noted that book interest is a significant predictor in obtaining books. In fact, as shown in the data, when in a wealthy family, students who reported no interest in reading are much less likely to purchase books than those who did. When compared to the likelihood of borrowing books and receiving books as gift, the probability of buying books is the most drastically affected by reading interest. This suggests that even among richer families, there is still a certain reticence to spend money on books, and an intrinsic motivation—personal interest in reading books—could cancel out this reluctance.

### 4.3. The Role of the Classroom Bookcase

We were also concerned about the evaluation of students on the public bookshelves of the classroom. Although, student should focus on genre of books to read, their reading interest may be negatively influenced once they feel uncomfortable to spend more time on reading books in their classroom. A diverse bookcase would be more exciting and might inspire students spend more time to read. As explained above, reading interest is tantamount to getting deeper knowledge and achieving better academic outcomes. Therefore, a bookcase with diverse books that appeals to varied interests in terms of topics would encourage students to spend more time on reading. Our regression results favor the idea that a bookcase with various types of books could increase the likelihood that students spend over 30 min per day for reading. In addition, it could once again be observed that while the effect of a satisfactory variety in the common bookcase enhanced the student's reading intensity in terms of duration, the most noticeable effects remained those of reading interest. This result complements the findings that the increase in reading resources improves the reading skills and children's attitudes towards reading [56,74,86]. As reading interest has been shown to be linked to personal preferences such as favorite leisure activities [107], diversifying both the classroom bookcase and the type of reading promotion activities would perhaps be a proposition worth considering. More focus on creating an encouraging reading environment, such as providing access to a wider variation of books in this case, would inspire students to read at higher intensity.

### 4.4. Occupational Aspiration and Parental Education

The question of whether or not having career orientation at the early stage of schooling helps improve the academic performance of students remains unclosed till today. In the literature of educational research, no examination is conducted with the correlation between career orientation of students and their academic performance. Most students participating in our survey reveal a concrete idea when asked about their dream future job. Only a small number of students do not think about their future career. We might think that job definition in the early stage of schooling motivates students to learn better therefore achieving academic outcomes. We have found a positive correlation between

"future job" and the average score of students' 45-min tests of math, physics, chemistry and biology. Evidence showed that students who define their job in the future are more likely to achieve better academic performance than those who have no career orientation, which is somewhat in line with the result found among 12th grade students in the US [97].

The role of parental involvement on the academic achievement of minority children is indicated in [120]. A family with higher education parents is aware of the benefits of reading and they tend to provide their children with more chance to read [90]. Our study revealed an evidence that the education level of fathers and mothers in families is significantly and postively associated with their children's academic outcome at schools. Most of students who were born and raised by highly educated parents ever experience knowledge and guidance from their families at their young ages [92,93]. Further, highly qualified parents are more likely to have better orientation for their children because they actually know what are good or not good thus giving good advice for the study of their children [94]. In the same vein, parents can act as professional instructors since high level of education comes with wisdom, parents help their children achieve academic success.

It should be worth noting that there seems to be a stronger involvement of the mother in the child's academic performance; as evidenced by estimate coefficients $\beta_{EduMot} = 0.721$ of mothers versus $\beta_{EduFat} = 0.306$ of fathers, when they have finished university or up. This might or might not be counter-intuitive to the reader, as certain cultures perhaps view the father more likely to assume the role of the educator. In the case of the Vietnamese household, however, this results seem to be in line with the traditional disparity in child-rearing roles between the mother and the father. Moreover, this finding might be explained that by the fact that, compared to fathers, mothers play a more important role in forming children's reading habits at home since they spend more time on reading, teaching and are more frequently the one to encourage their children to read [57,91]. As the study has been conducted on adolescents aged 11 to 15, considered "young", their education more often fall under the domain of the mother, in conventional views. As such, the notable difference in influence between the two parent here suggests a still-present division of responsibility. Literacy promotion initatives should take into account this difference, both to enhance the effectivity of reading promotion activities and to tackle this gender gap in child education.

## 5. Conclusions

The article investigated the association between Vietnamese students' STEM-related academic achievement and their reading interest, living condition, parental education, and their aspiration for future career. Reading interest, a defined occupational aspiration, the education level of the mother are predictors of students' higher score in STEM-related subject, namely Math, Physics, Chemistry and Biology. Regarding the types of books, students who read natural sciences books achieved higher academic results than students who preferred other topics. Reading interest also predicts the book purchasing behavior of the students, even in a wealthy family.

Vietnam is aiming to achieve United Nations' Sustainable Development Goal by 2030, with one of the top priorities is ensuring the goal of SDG 4, which aims for inclusive, equitable and quality education and for promoting lifelong learning opportunities for all [121,122]. In order to achieve this goal, STEM education is being considered as a practical and holistic education approach [123,124]. The research results provide educational policy makers in Vietnam strong empirical evidences regarding reading interest and STEM education of Vietnamese students [125]. The effect of reading topic-specific books materializes into students' academic achievement, as our results suggested. Thus, fostering the interest for reading in students should be the focus of school's curriculum or education policy. The government and the school in Vietnam can do this by designing a more reading-friendly environment and allowing students more time to read, as these were suggested to correlate with higher academic achievement. Finally, the involvement of student's family is crucial to fostering reading interest and improving student academic achievement, especially the role of the mother.

Despite the contribution to education policy in Vietnam, this study is still subjected to certain limitations. Firstly, the study was conducted in a Northern city in Vietnam, thus any generalization to wider population must be cautious. Secondly, the study employed the frequentist methods, which still have weaknesses such as the definition of statistical significance [126]; thus, replications or further studies can use different approaches such as qualitative technique, which would help triangulate the quantitative data analysis, or Bayesian statistics, which strength lies in the ability to update probability with new arising evidences [127].

**Author Contributions:** Conceptualization, T.-T.V. and Q.-H.V.; Data curation, T.-P.-T.T., T.-P.-T.N., H.-M.V. and M.-T.H.; Formal analysis, T.-T.-H.L., T.T., T.-P.-T.N., T.-H.V. and M.-H.N.; Methodology, T.-T.V. and Q.-H.V.; Project administration, T.T. and Q.-H.V.; Supervision, T.T. and Q.-H.V.; Validation, T.-P.-T.T., C.-T.N., H.-M.V., M.-H.N. and M.-T.H.; Visualization, H.-M.V. and M.-H.N.; Writing—Original draft, T.-T.V., T.-H.V., D.-Q.B. and P.-H.H.; Writing—Review & editing, T.-T.-H.L., C.-T.N., T.-T.V., T.-H.V., D.-Q.B., P.-H.H. and M.-T.H.

**Funding:** This research received no external funding.

**Acknowledgments:** We would like to send our gratitude to research staff of Vuong & Associates (Hanoi, Vietnam) for assisting in collecting data, especially Do Thu Hang, and Dam Thu Ha. Our most sincere thanks also go on to personnel of junior high schools and provincial departments that provided support during the survey.

**Conflicts of Interest:** The authors declare no conflict of interest.

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
