# Peer review of "Reading Habits, Socioeconomic Conditions, Occupational Aspiration and Academic Achievement in Vietnamese Junior High School Students"

_sustainability, doi:10.3390/su11185113_

Round 1

Reviewer 1 Report

The authors explore students’ academic achievement, reading passion, family socioeconomic condition, parental education and occupational aspiration.

The manuscript is well-written, so it is comprehensible.

Some issues to consider:

Abstract: It would be a good idea to add the complete term for “SDG4” Literature review. When authors explain different studies from other different authors, it would be a good idea to add the country of the samples and the age of the children (not only the grade) to give more information to the readers. Page 6, line 285. The title “Parents’ education and career and students’ academic performance” should add “and occupational aspiration” because the section is about occupational aspiration too. 1.4. Research questions and hypothesis. There is no research question related to parental education. Authors should clarify the research question related to this issue considering that in the abstract authors speak about this issue (“…explore students’ academic achievement and its association with their reading passion, family socioeconomic condition, parental education…”). 1.4. Research questions and hypothesis. H1 (lines 390-391). It would be a good idea to clarify the hypothesis in terms of the idea of “positively and negatively correlated”. 2.2.1. Dataset. It would be a good idea to add the number of questions/items of the survey. Also, how much time students used to answer the survey? Was it during class time? 2.2.2. Variables. Authors should explain a little more about the differences between natural sciences books and social sciences and humanities books, to clarify what types of books are each one. 3.1. Descriptive statistics. Considering that this journal is an international journal, it would be a good idea to specify the age of the students in each grade (grade 6, grade 7, grade 8, grade 9) to clarify the characteristics of the sample to international readers. Table 1. There is no row for “30 minutes or over” for the variable “TimeSoc”. Table 1. Regarding the variables “Edufat” and “Edumot”, the frequency data are not equal to the complete sample. Were there students who did not answer these questions? Figure 1. Authors should explain the figure in the main text. Also, the authors should add a table with the information of this variable (APS45), indicating the minimum value, maximum value, the average and standard deviation of the sample. 3.2.1. Authors should use the same research questions in the section related to research questions (page 8, lines 377-387) and in the section in which they explain the results of each research question (pages 12-19). Line 793. Authors say “That said, it might also very well be the case that   As such, the notable…”. This sentence is not clear. It would be a good idea to go in depth in the discussion, comparing the results with more authors (for example, from the literature review). Authors should add more information about the implications of the results concerning SDG4, considering that in the abstract they say “The results present implications for education policy making with a vision towards SDG4”. It would be a good idea to add more information about the limitations of this study and further research.

Author Response

Dear reviewer 1,

Once again we would like to express our most sincere thanks for your contribution in helping us improve our submission. We have done major revisions following your very detailed input. The changes brought to the manuscript have been highlighted in yellow, whereas newly added passages were highlighted in green. In this letter, we will address each of the points you have made individually.

The authors explore students’ academic achievement, reading passion, family socioeconomic condition, parental education and occupational aspiration. The manuscript is well-written, so it is comprehensible.

Thank you for your compliment.

Some issues to consider: Abstract: It would be a good idea to add the complete term for “SDG4”

We have added the complete term for “SDG4” accordingly.

Literature review. When authors explain different studies from other different authors, it would be a good idea to add the country of the samples and the age of the children (not only the grade) to give more information to the readers.

We have provided more information to the readers in terms of country of the samples or the age of the children.

Page 6, line 285. The title “Parents’ education and career and students’ academic performance” should add “and occupational aspiration” because the section is about occupational aspiration too.

We have added “and occupational aspiration” to the title in Page 6 accordingly.

4. Research questions and hypothesis. There is no research question related to parental education. Authors should clarify the research question related to this issue considering that in the abstract authors speak about this issue (“…explore students’ academic achievement and its association with their reading passion, family socioeconomic condition, parental education…”).

The association between parental education and students’ academic achievement was explored in RQ5. We have clarified the research question to make it clearer.

4. Research questions and hypothesis. H1 (lines 390-391). It would be a good idea to clarify the hypothesis in terms of the idea of “positively and negatively correlated”.

We have clarified Hypothesis 1:

H1: Reading habits and interest in natural sciences books positively predict the average score of students' 45-minute tests of Math, Physics, Chemistry and Biology

2.1. Dataset. It would be a good idea to add the number of questions/items of the survey. Also, how much time students used to answer the survey? Was it during class time?

We have added the number of questions/items of the survey. The survey was carried out by the homeroom teachers at junior high schools in the Ninh Binh province. The questionnaire was thoroughly explained to the students by their teachers to ensure accuracy and validity of the records.

2.2. Variables. Authors should explain a little more about the differences between natural sciences books and social sciences and humanities books, to clarify what types of books are each one.

We have explained the different between natural sciences books and social sciences and humanities books and why certain types of books are grouped into each one:

-           Typebookgr: recoded variable from Typebook, grouping ‘a’, ‘b’ and ‘d’ into ‘gr1’ and ‘c’, ‘e’ and ‘f’ into ‘gr2’. Group 1 as represented by ‘gr1’ consists of novels, biographies and arts, which are characterized by the story of human and human relationships. This group of book type is either fiction books or social sciences and humanities books. Group 2 as represented by ‘gr2’ is comprised of the genres popular science and vocational instruction (and others), which could be considered non-fiction or natural science books because they mainly focus on practical knowledge and vocational skills;

1. Descriptive statistics. Considering that this journal is an international journal, it would be a good idea to specify the age of the students in each grade (grade 6, grade 7, grade 8, grade 9) to clarify the characteristics of the sample to international readers.

We have clarified this information in the 2.2.1. Dataset section:

Junior high school students, aged 11 to 15 (from grade 6 to grade 9), were randomly selected to respond to the questionnaires directly, in written form, after having received thorough explanations from their instructors to ensure accuracy and validity of the records

Table 1. There is no row for “30 minutes or over” for the variable “TimeSoc”.

We have added a new row for “30 minutes or over” for the variable “TimeSoc”

Table 1. Regarding the variables “Edufat” and “Edumot”, the frequency data are not equal to the complete sample. Were there students who did not answer these questions?

There were students who did not answer these questions because of the family situations. For instance, a student could be raised by his/her grandparents because his/her parents passed away.

Figure 1. Authors should explain the figure in the main text. Also, the authors should add a table with the information of this variable (APS45), indicating the minimum value, maximum value, the average and standard deviation of the sample.

We have provided a table with the information for variable ‘APS45’ and explained Figure 1 in the main text.

2.1. Authors should use the same research questions in the section related to research questions (page 8, lines 377-387) and in the section in which they explain the results of each research question (pages 12-19).

Previously, we adjusted the wording in some research questions, which made the research questions in two sections 1.4. Research questions and hypothesis and 3.2. Regression results appeared different. Thus We have made the research questions in both sections similar to avoid any possible confusion.

Line 793. Authors say “That said, it might also very well be the case that As such, the notable…”. This sentence is not clear.

We have edited the sentence to make it clearer.

It would be a good idea to go in depth in the discussion, comparing the results with more authors (for example, from the literature review). Authors should add more information about the implications of the results concerning SDG4, considering that in the abstract they say “The results present implications for education policy making with a vision towards SDG4”. It would be a good idea to add more information about the limitations of this study and further research.

A new section: 5. Conclusion was added to provide an in-depth discussion of our findings:

Conclusion

The article investigated the association between Vietnamese students’ STEM-related academic achievement and their reading interest, living condition, parental education, and their aspiration for future career. Reading interest, a defined occupational aspiration, the education level of the mother are predictors of students’ higher score in STEM-related subject, namely Math, Physics, Chemistry and Biology. Regarding the types of books, students who read natural sciences books achieved higher academic results than students who preferred other topics. Reading interest also predicts the book purchasing behavior of the students, even in a wealthy family.

Vietnam is aiming to achieve United Nations’ Sustainable Development Goal by 2030, with one of the top priorities is ensuring the goal of SDG 4, which aims for inclusive, equitable and quality education and for promoting lifelong learning opportunities for all [122,123]. In order to achieve this goal, STEM education is being considered as a practical and holistic education approach [124,125]. The research results provide educational policy makers in Vietnam strong empirical evidences regarding reading interest and STEM education of Vietnamese students [126]. The effect of reading topic-specific books materializes into students’ academic achievement, as our results suggested. Thus, fostering the interest for reading in students should be the focus of school’s curriculum or education policy. The government and the school in Vietnam can do this by designing a more reading-friendly environment and allowing students more time to read, as these were suggested to correlate with higher academic achievement. Finally, the involvement of student’s family is crucial to fostering reading interest and improving student academic achievement, especially the role of the mother.

Despite the contribution to education policy in Vietnam, this study is still subjected to certain limitations. Firstly, the study was conducted in a Northern city in Vietnam, thus any generalization to wider population must be cautious. Secondly, the study employed the frequentist methods, which still have weaknesses such as the definition of statistical significance [127]; thus, replications or further studies can use different approaches such as qualitative technique, which would help triangulate the quantitative data analysis, or Bayesian statistics, which strength lies in the ability to update probability with new arising evidences [128].

We believe that we have put an appropriate amount of efforts in responding to all of your concerns. We hope that you find our revision satisfactory, and would like to once again thank you for your suggestions. We are truly honored to be able to work with you in improving this manuscript.

With our highest respects,

The authors

Reviewer 2 Report

Well done to the team of researchers.  

This research study is illuminating  as despite commonplace assumptions that teenagers do not read in this digital age, the survey findings show that Vietnamese  teenagers  still enjoy reading for pleasure and that having access to scientific  texts raises  achievement  in  STEM subjects. 

 The article is well structured with a clear abstract and introduction which enable the reader to quickly understand the research context. Extensive literature review provides good support  from a wide range of relevant academic literature.  This helps to place the study into context for the reader. 

All the research processes are considered well and the findings are linked clearly to the research questions through the use of description, tables and charts. 

The discussion is thorough and analyses the findings well.  A minor point but I would have liked to have seen final thoughts on any implications of the findings  for future practice and policy. 

The article is written well but there are a few corrections to be made: 

Line 121: A reference is needed to support statement  that Vietnam is a lower middle  income country otherwise it is just conjecture 

Line 186 : replace  to read with reading 

Line 188 Replace England with English 

The discussion regarding parental  involvement  particularly involvement of the mother in the child’s academic performance would be strengthened by links to literature. 

Author Response

Dear reviewer 2,

Once again we would like to express our most sincere thanks for your contribution in helping us improve our submission. We have done major revisions following your very detailed input. The changes brought to the manuscript have been highlighted in yellow, whereas newly added passages were highlighted in green. In this letter, we will address each of the points you have made individually.

Well done to the team of researchers. This research study is illuminating as despite commonplace assumptions that teenagers do not read in this digital age, the survey findings show that Vietnamese teenagers still enjoy reading for pleasure and that having access to scientific texts raises achievement in STEM subjects. The article is well structured with a clear abstract and introduction which enable the reader to quickly understand the research context. Extensive literature review provides good support from a wide range of relevant academic literature. This helps to place the study into context for the reader. All the research processes are considered well and the findings are linked clearly to the research questions through the use of description, tables and charts.

Thank you for your comments. These encouraging words truly give us the motivation to continue our works, to keep on finding new and interesting results, and to contribute to the betterment of the world.

The discussion is thorough and analyses the findings well. A minor point but I would have liked to have seen final thoughts on any implications of the findings for future practice and policy.

We have added a new section 5. Conclusion to provide final thoughts on the implications of the findings, suggestions for future practice and policy, and discussion of potential limitations of the study.

The article is written well but there are a few corrections to be made: Line 121: A reference is needed to support statement that Vietnam is a lower middle income country otherwise it is just conjecture Line 186: replace to read with reading Line 188 Replace England with English

We have edited the article according to your suggestions.

The discussion regarding parental involvement particularly involvement of the mother in the child’s academic performance would be strengthened by links to literature.

We have linked the discussion regarding parental involvement particularly the involvement of the mother in the child’s academic performance to the literature, mostly from the Literature Review.

We believe that we have put an appropriate amount of efforts in responding to all of your concerns. We hope that you find our revision satisfactory, and would like to once again thank you for your suggestions. We are truly honored to be able to work with you in improving this manuscript.

With our highest respects,

The authors

Reviewer 3 Report

The relevance of the research problem, the appropriate methodological application and the validity of the results obtained guarantee the quality of the manuscript.

However, it is suggested to incorporate a brief section that includes the limitations of the research, such as the absence of application of qualitative research techniques, which would help triangulate the quantitative data analyzed.

Author Response

Dear reviewer 3,

Once again we would like to express our most sincere thanks for your contribution in helping us improve our submission. We have done major revisions following your very detailed input. The changes brought to the manuscript have been highlighted in yellow, whereas newly added passages were highlighted in green. In this letter, we will address each of the points you have made individually.

The relevance of the research problem, the appropriate methodological application and the validity of the results obtained guarantee the quality of the manuscript.

Thank you for your comments. These encouraging words truly give us the motivation to continue our works, to keep on finding new and interesting results, and to contribute to the betterment of the world.

However, it is suggested to incorporate a brief section that includes the limitations of the research, such as the absence of application of qualitative research techniques, which would help triangulate the quantitative data analyzed.

Thank you for your suggestion. We have added a new section: 5. Conclusion, to discuss the limitations of the research, as well as the implications of the findings to future educational policy.

We believe that we have put an appropriate amount of efforts in responding to all of your concerns. We hope that you find our revision satisfactory, and would like to once again thank you for your suggestions. We are truly honored to be able to work with you in improving this manuscript.

With our highest respects,

The authors